# Transcriptome Profiling of Potato (*Solanum tuberosum* L.) Responses to Root-Knot Nematode (*Meloidogyne javanica*) Infestation during A Compatible Interaction

**DOI:** 10.3390/microorganisms8091443

**Published:** 2020-09-21

**Authors:** Teresia N. Macharia, Daniel Bellieny-Rabelo, Lucy N. Moleleki

**Affiliations:** Department of Biochemistry, Genetics and Microbiology, Forestry and Agricultural Biotechnology Institute, University of Pretoria, Pretoria 0002, South Africa; tessyn.nyambura28@gmail.com (T.N.M.); daniel.bellieny@gmail.com (D.B.-R.)

**Keywords:** root-knot nematode, *Solanum tuberosum*, differentially expressed genes, defense response

## Abstract

Root-knot nematode (RKN) *Meloidogyne javanica* presents a great challenge to Solanaceae crops, including potato. In this study, we investigated transcriptional responses of potato roots during a compatible interaction with *M. javanica*. In this respect, differential gene expression of *Solanum tuberosum* cultivar (cv.) Mondial challenged with *M. javanica* at 0, 3 and 7 days post-inoculation (dpi) was profiled. In total, 4948 and 4484 genes were detected, respectively, as differentially expressed genes (DEGs) at 3 and 7 dpi. Functional annotation revealed that genes associated with metabolic processes were enriched, suggesting they might have an important role in *M. javanica* disease development. MapMan analysis revealed down-regulation of genes associated with pathogen perception and signaling suggesting interference with plant immunity system. Notably, delayed activation of pathogenesis-related genes, down-regulation of disease resistance genes, and activation of host antioxidant system contributed to a susceptible response. Nematode infestation suppressed ethylene (ET) and jasmonic acid (JA) signaling pathway hindering JA/ET responsive genes associated with defense. Genes related to cell wall modification were differentially regulated while transport-related genes were up-regulated, facilitating the formation of nematode feeding sites (NFSs). Several families of transcription factors (TFs) were differentially regulated by *M. javanica* infestation. Suggesting that TFs play an indispensable role in physiological adaptation for successful *M. javanica* disease development. This genome-wide analysis reveals the molecular regulatory networks in potato roots which are potentially manipulated by *M. javanica*. Being the first study analyzing transcriptome profiling of *M. javanica*-diseased potato, it provides unparalleled insight into the mechanism underlying disease development.

## 1. Introduction

Potato, *Solanum tuberosum* (L), belongs to the Solanaceae family, which comprises several economically important crops such as tomato, pepper, aubergine, and tobacco. Plant parasitic nematodes, particularly root-knot nematodes (RKNs) are among the most destructive and economically important pests of potatoes worldwide [1,2]. In this context, *Meloidogyne* spp. are obligate and highly polyphagous pests that form an intricate relationship with their host, causing drastic morphological and physiological changes in plant cell architecture [3].

A typical life cycle of RKNs spans between 4–6 weeks depending on the nematode species and environmental conditions. Following the embryonic phase, the infective second-stage juveniles (J2s) hatch from the egg. At 3 dpi, the nematodes have already penetrated the host root tips and migrating towards the elongation zone [4]. At this stage, the J2s select target cells to initiate reprogramming of host cells to giant cells (GCs). The nematodes are completely dependent on the induced GCs for their supply of nutrients. During the induction stage, the parasitic J2 abandons its migratory lifestyle, becomes sedentary to concentrate on feeding, development, and reproduction [5]. As the GCs enlarge, surrounding cells undergo rapid division causing swelling of roots and discontinuity of the vascular tissue. The sedentary nematode further molts into J3 and J4 stages, and finally into the adult stage when the nutrient acquisition stage commences from 7 dpi onwards. At the nutrient acquisition, the developing nematode, GCs, and surrounding tissue all contribute to the formation of typical RKN symptoms. These symptoms resemble large tumor-like galls that are readily visible to the naked eye [5]. Analogous to other plant pathogens, nematode secretions play a crucial role in manipulation of the host cellular function. Secreted molecules suppress host defense to initiate a successful infection process, including the establishment and maintenance of NFSs [6]. In the genus *Meloidogyne*, several effectors have been reported such as MiLSE5, which interferes with host metabolic and signaling pathways, or MjTTL5, Misp12, and MgGPP, which suppress the host immune responses facilitating successful nematode parasitism [7].

Due to their capability to infest plant species from diverse families, RKN species pose a great challenge to crop production globally [8]. In 2014, 22 species of RKN were reported in Africa causing damage to various vegetable and field crops. For decades, the use of nematicides has been effective in managing RKN populations. However, their usage is coupled with adverse effects on the ecosystem. This has led to the withdrawal of the most effective nematicides from the agro-markets, further aggravating crop losses due to RKN [9]. Plant host resistance through the use of resistant cultivars is an effective and environmentally safe alternative method of controlling RKN species [9]. Nevertheless, the current cultivated potato cultivar lacks resistance against RKN [10]. Thus, studies involving plant–nematode interactions will deepen our understanding of the molecular regulatory networks associated with resistance or susceptibility. The insights drawn from such studies will be useful in breeding programs to develop novel target-specific control strategies against nematodes.

RNA-Sequencing has become a powerful instrument for gene expression profiling and detection of novel genes [11,12] which has been used widely to study the expression profiles of RKN diseased Solanaceae plants [13,14]. RNA-seq profiling has been used to decipher potato responses to various abiotic [15,16] and biotic stresses [17,18] where large sets of genes and pathways associated with either biotic and abiotic stress were revealed. To date, most research has focused on potato gene expression in response to potato cyst nematodes [19,20,21] while potato responses to RKN infestation remain poorly understood. Here, we set out to evaluate the molecular basis of Potato-RKN compatible interaction, we employed RNA-Seq to analyze differential gene expression patterns in *S. tuberosum* cv. Mondial subjected to *M. javanica* infestation at two early stages (3 and 7 dpi).

## 2. Materials and Methods

### 2.1. Plant Material and Meloidogyne javanica Inoculations

Certified potato seed (tubers) of seven South African commercial potato cultivars were evaluated for resistance to *M. javanica* under greenhouse conditions. The seed tubers were pre-germinated in the dark 20 ± 3 °C for seven days to allow sprouting. Stocks of *M. javanica* were originally obtained from Dr. Pofu (ARC Roodeplaat, Pretoria, South Africa) and maintained on a susceptible tomato cultivar, *S. lycorpersicum* cv. Floradade in the glasshouse at 24–30 °C, for eight weeks. *M. javanica* eggs were extracted from infested roots as described [22]. Egg suspension was poured onto an extraction tray for the collection of second-stage juvenile (J2s) nematodes. Five-week-old potato seedlings were inoculated with 1000 freshly hatched J2s per plant and control plants mock-inoculated with sterile water. Taylor et al. [23] ranking scale was used to determine susceptibility while Sasser et al. [24] reproduction factor (RF) formula was used to assess the host status of potato cultivars. For RNA experiments, whole root tissues of a compatible potato cultivar were collected at 0, 3, and 7 days post-inoculation (dpi) with two biological replicates per time point. Samples were washed and immediately frozen in liquid nitrogen to prevent RNA degradation and later stored at −80 °C until RNA extraction.

### 2.2. RNA Extraction, Library Preparation, and Sequencing

RNA extraction, library preparation, and sequencing were carried out at Novogene (HK) Company Limited. Total RNA for individual time course and replicates was extracted using TiaGen extraction kit (Biotech Beijing Co., Ltd., Beijing, China) and treated with sigma DNase1 (D5025). RNA degradation and contamination was measured on 1% agarose gel while RNA purity was assessed using the NanoPhotometer^®^ spectrophotometer (IMPLEN, Westlake Village, CA, USA). RNA concentration and integrity were assessed using Qubit^®^ RNA Assay kit in Qubit^®^ 2.0 Fluorometer (Life Technologies, Carlsbad, CA, USA) and RNA Nano 6000 Assay Kit of the Bioanalyzer 2100 system (Agilent Technologies, Santa Clara, CA, USA), respectively. Three micrograms of RNA samples were used as input for library construction. Libraries were constructed using NEBNext^®^ Ultra™ RNA Library Prep Kit for Illumina^®^ (NEB, Ipswich, MA, USA) according to the manufacturer’s instructions and index codes were added to attribute sequences to each sample. Finally, PCR products were purified using AMPure XP system and quality of the library assessed using the Agilent Bioanalyzer 2100 system. A cBot Cluster Generation System using HiSeq PE Cluster Kit cBot-HS (Illumina) was used to cluster the index-coded samples. After cluster generation, the library preparations were sequenced on an Illumina Hiseq platform 2500 generating 150-bp paired-end reads.

### 2.3. Transcriptomic Data Analysis

Quality analysis of sequenced reads were initially analyzed using FASTQC package (https://www.bioinformatics.babraham.ac.uk/projects/fastqc). Clean reads were obtained by removing reads containing adapter reads with poly-N and low-quality reads from raw data. Trimming of low-quality regions was performed using Trimmomatic v 0.36 [25]. All the subsequent downstream analyses were based on high-quality data. *Solanum tuberosum* genome v4.03 [26] was used for reference-guided mapping of RNA-seq reads. Paired-end clean reads were aligned to the potato genome using hisat2 v 2.1.0 software [27]. Unmapped reads were progressively trimmed at the 3′end and re-mapped to the genome. Next, featureCounts package [28] was used to perform raw-reads counts in R environment (https://www.r-project.org/). The read counts were then used for differential expression analysis using edgeR package [29]. Further, to investigate the responses at different time points (3 dpi and 7 dpi), the expression profiles were compared to mock-inoculated (0 dpi) data sets. The transcripts were then classified as differentially expressed genes (DEGs) based on both (a) false discovery rate (FDR) [30] cut-off of 0.05 and (b) log_2_ fold change ≥ 1 or ≤−1 for induced and repressed genes, respectively.

### 2.4. Gene Ontology (GO) and Enrichment Analysis

The GO and enrichment analysis were performed using agriGO v.2.0 [31] and categorized by WEGO v 2.0 tool [32]. Parametric gene set enrichment analysis based on differential expression levels (log_2_ fold change) was performed and FDR correction was performed using the default parameters to adjust the *p*-value. Functional annotations and pathway analyses were obtained through sequence search performed on eggNOG database utilizing *eggmapper* [33]. Annotations from eggNOG were then integrated with Kyoto Encyclopedia of Genes and Genomes (KEGG) database to reach pathway annotation level. In addition, we used MapMan software to visualize the biotic stress pathway to identify the genes that are known to be part of the cascade of defense signals in response to pathogen or pest invasion [34].

### 2.5. Validation for DEGs by qRT-PCR

For qRT-PCR, first-strand cDNA was done from total RNA using Superscript IV First-Strand cDNA Synthesis SuperMix Kit (Invitrogen, Carlsbad, CA, USA) following manufacturer’s protocol. Quantitative real-time PCR was performed using SYBR Green Master Mix in the QuantStudio 12k Flex Real-Time PCR system (Life Technologies, Carlsbad, CA, USA) to validate DEGs. Two micrograms of the sample (concentration) were added to 5 μL of Applied Biosystems SYBR Green Master Mix and primers at a concentration of 0.4 μM. The amplification cycle consisted of the following: initial denaturation at 50 °C for 5 min and 95 °C for 2 min followed by 45 cycles of 95 °C for 15 s and 60 °C for 60 s. Each sample was run in triplicates. Specific qRT-PCR primers for six target genes were designed using an online tool Prime-Blast (http://www.ncbi.nlm.nih.gov/tools/primer-blast) (Appendix A). Each sample was run in triplicates. The 18S rRNA and elongation factor 1-α (PGSC0003DMG400020772,ef1α), [35] were used as the reference genes for normalization, and the mock-treated samples used as calibrators. The comparative 2^−∆∆*Ct*^ method was used to determine the relative fold change according to Schmittgen and Livak [36]. Despite the two techniques (RNA-seq and qRT-PCR) being different, the expression patterns of selected genes upon nematode infestation was consistent between the two procedures (Appendix A).

### 2.6. Data Access

Both raw and processed sequencing data have been deposited to the Gene Expression Omnibus (GEO) repository at the National Center for Biotechnology Information (NCBI) with accession no. GSE134790.

### 2.7. Statistical Analyses

To identify differentially expressed genes with significant differences between the control (0 dpi) and 3 and 7 dpi infestation stages, statistical analyses were performed in R environment using edgeR package. The *p*-values were corrected using the Benjamini and Yekutieli [30] approach.

## 3. Results and Discussion

### 3.1. Analysis of Meloidogyne javanica Infestation and Functional Annotation of Differentially Expressed Genes

To establish whether *Solanum tuberosum* cv. Mondial is susceptible or tolerant to *M. javanica*, five-week-old potato plants were inoculated with 1000 J2s eggs and the control mock-inoculated with sterile water. At eight weeks post-inoculation, the results showed that *S. tuberosum* cv. Mondial plants infested with *M. javanica* have a reproduction factor (Rf > 1) of 3.11 and galling index of 5 (Figure 1A).

Based on the GI (>100 per root system) *S. tuberosum* cv. Mondial potato cultivar was classified as highly susceptible. Figure 1B,C illustrate mature galls at eight weeks after inoculation exhibited either as single or egg masses, confirming *M. javanica* infectivity and ability to reproduce and complete their life cycle on this potato cultivar.

To understand the molecular basis of this compatible interaction between *M. javanica* and *S. tuberosum* cv. Mondial, RNA sequencing was performed at two infestation stages: 3 and 7 dpi. These time points correspond to nematode stages of induction of feeding sites at 3 dpi and nutrient acquisition stage that starts from 7 dpi to 8 weeks after infestation [5]. Approximately 1.3 billion paired-end reads were generated yielding an average of 23 million high-quality reads for individual samples. Successfully mapped reads onto the *S. tuberosum* reference genome (v4.03) [26] accounted for 78–86% of the total number of reads generated per sample (Appendix A). Log_2_ fold change ≥ ±1 and adjusted *p*-value (FDR) < 0.05 were used as cut off values to obtain DEGs through pairwise comparison between the mock-inoculated and infested samples at 3 and 7 dpi. Overall, 4948 genes were differentially expressed at 3 dpi. Of these, 2867 were down-regulated and 2081 up-regulated. At 7 dpi, fewer genes (4484) were differentially expressed compared to 3 dpi. The number of down-regulated genes at 7 dpi (2871) was similar to that observed at 3 dpi (2867); however, there were 22% fewer genes up-regulated at 7 dpi compared to those at 3 dpi (Figure 2A and Appendix A).

Collectively, 3108 genes were differentially regulated at both 3 and 7 dpi: 2069 down- and 1022 up-regulated. For both time points, a total of 3652 out of 6324 (57.75%) of the DEGs were down-regulated compared to only 42.25% which were up-regulated (Figure 2A,B and Appendix A). One possible explanation of this scenario is that down-regulation of genes might be essential for proper formation of galls induced by RKN as reported previously by Jammes et al. [37]. Secondly, as an obligate biotroph, *M. javanica* establishes an intricate relationship with its host; therefore, a larger suppression of defense-related genes is also plausible.

GO enrichment analyses revealed the main regulatory trends in root tissues in response to *M. javanica* infestation. The GO terms were grouped into three main functional categories at adjusted *p*-value < 0.05 and categorized using WEGO software [32]. Within the biological process class, the highest percentage of the DEGs was down-regulated and fell under metabolic process category. Within this category, we found the following sub-categories: Primary and cellular metabolic process, biosynthetic and oxidation-reduction process as well as regulation of metabolic processes. Other significant GO terms in this category include response to stimulus, cellular process, localization and signalling processes, and regulation of biological processes (Figure 3 and Appendix A).

Further, MapMan analysis using the biotic stress overview classified potato genes putatively involved in mediating responses to nematode attack. A total of 1471 and 1408 genes at 3 and 7 dpi, respectively, belonging to hormone signaling, cell wall organization, proteolysis, redox homeostasis, signaling, transcriptional regulation as well as defense-related genes (secondary metabolites, PR proteins, heat shock proteins, and proteinase inhibitors) were found to be under differential regulation following *M. javanica* infestation (Figure 4A,B).

Genes related to redox state, signaling, beta-glucanase and cell wall displayed similar expression patterns at both stages 3 and 7 dpi. Genes encoding for peroxidases, ERF, and WRKY were down-regulated mostly at 7 dpi compared to the 3-dpi infestation stage. Genes related to secondary metabolism and proteolysis exhibited a general up-regulation at 3 dpi in comparison to 7dpi (Figure 4A,B).

### 3.2. Pathogen Perception and Regulation of Defense Response Genes by Meloidogyne javanica Infestation

Plant recognition of *M. javanica* presence, migration, and feeding is the initial step in triggering activation of downstream signaling cascades to induce defense responses. Large arsenals of plant receptors are responsible for recognition of phytonematodes and subsequent induction of basal plant defenses [38,39]. Eight genes encoding for membrane-localized proteins, *mildew resistance locus o* (*MLO*) family protein were down-regulated at 3 and/or 7 dpi (Figure 5, Appendix A).

Additionally, an important group of pattern recognition receptors (PRRs) was differentially regulated with the highest proportion down-regulated (62.86% of 175 genes). These include leucine-rich repeat receptor-like protein kinase (LRR-RLK), receptor-like protein kinases (RLP), and wall-associated receptor kinases (WAKs) (Appendix A) indicating the importance of PRRs in mediating plant immunity to *M. javanica.* Among the down-regulated LRR-RLK genes were two genes encoding for BAK1-interacting receptor-like kinase 1. Further, we found an LRR-RLK encoding gene, polygalacturonase inhibiting protein 2 (*PGIP 2*) down-regulated 7-fold at 3 dpi and 4-fold at 7 dpi (Figure 5 and Appendix A). In addition to suppression of a cell wall-associated kinase, plant elicitor peptides (*PEP1* receptor, three genes) encoding genes that perceive damage-associated molecular proteins (DAMPs) were down-regulated at both 3 and 7 dpi (Figure 5 and Appendix A). These results suggest that *M. javanica* likely interferes with DAMP-mediated immunity subduing activation defense response.

Transmission of perceived signals from the PRRs is mediated through the MAPK cascade and calcium (Ca^2+^) signaling pathway which transfers downstream components of plant immunity. In this regard, we found the expression of MAPKs genes was largely repressed (9 out of 11 genes) by nematode infestation (Figure 5 and Appendix A). Furthermore, *M. javanica* repressed 88% of the genes involved in Ca^2+^ signaling pathway (66 out of 75 genes) (Appendix A).

The intracellular nucleotide-binding domain leucine-rich repeat (NBS-LRR) proteins are important in recognizing PPNs effectors leading to effector-triggered immunity. The majority of these NBS-LRR proteins are a major class of disease resistance (R) proteins encoding resistance R-genes [39]. In the current study, we identified 32 genes encoding for either toll-interleukin 1 receptor (TIR) domain or coiled-coil (CC), which are the two subgroups of NBS-LRR proteins. Of these, 20 genes were down-regulated at 3 and/or 7 dpi following nematode infestation interfering with the host immune response (Figure 5 and Appendix A).

PTI activation is associated with expression of pathogenesis-related (PR) proteins which defense-related genes expressed upon infection by diverse pathogens including nematode pests. These PR proteins contribute to systemic acquired resistance (SAR) and are induced through the action of signaling compounds such as SA, JA, and ET [40]. Generally, we detected a delayed activation of the PR genes as the majority of the PR encoding genes were up-regulated at 7 dpi (Figure 5 and Appendix A). This could reflect a strategy adopted by the *M. javanica* to suppress defense response associated with PR proteins at early stages of invasion to ensure successful nematode infection. However, the induction of these PR proteins at 7dpi could have other roles in mediating a successful *M. javanica* infestation. For instance, we identified nine genes coding osmotin34 (PR-5), where 7 genes were up-regulated at 7 dpi (Figure 5 and Appendix A). Additionally, 11 genes encoding basic chitinase and chitinase-like family protein (PR-3 and PR-4) were detected with nine genes up-regulated at 7dpi (Figure 5 and Appendix A).

Rapid generation of reactive oxygen species (ROS) is one of the early PTI cellular events that trigger several defense responses such as activation of several defense genes and cell wall reinforcement [39]. In this study, respiratory burst homolog (RBOHs) and Riboflavin synthase-like superfamily which are important players in production of ROS in plants, were down-regulated both at 3 and/or 7 dpi (Figure 5 and Appendix A). In a resistant tomato genotype, ROS was strongly induced in response to RKN infestation mediating the R-gene resistant response [41]. Moreover, genes coding for peroxidases (50 genes) were detected mostly down-regulated (70%) during disease development. Emerging evidence shows that RKN can utilize the host ROS scavenging system to reduce the damaging effects of oxygen species [42,43]. Here, we detected one gene encoding for peroxiredoxin (PGSC0003DMG401002721), the main detoxifying antioxidant enzyme in the plant–nematode interface [44], being up-regulated at both time points. Glutathione S transferase (GSTs) encoding genes were also identified in this study. The majority of these were up-regulated at both time points (of these, 19 genes out of 25 were up-regulated (Figure 5 and Appendix A).

The production of anti-nematode compounds in the form of secondary metabolites play a critical role in induced plant immune responses against PPNs [39]. Analysis of genes related to secondary metabolism alterations was revealed in phenlyprononaids, lignin, flavonoids, isoprenoid, and phenols biosynthetic pathways (Appendix A). The three genes encoding for key enzymes (phenylalanine ammonia-lyase (*PAL*), cinnamate 4-hydroxylase (*CH4*), and 4-coumarate CoA-ligase 2 (*4CL*) essential for the synthesis of phenylpropanoids and downstream synthesis of other metabolites such as flavonoids, lignin, and SA [45] were down-regulated in this study (Appendix A). Other downstream genes detected to be differentially regulated include chalcone synthase, chalcone-flavone, and cinnamyl alcohols dehydrogenase involved in flavonoids and lignin pathway (Appendix A). This can imply that nematode interacts with phenylpropanoid metabolism to interfere with defense compounds that originate from this biosynthetic pathway.

Further, genes encoding for enzymes involved in isoprenoid biosynthetic pathway such as hydroxymethylglutaryl-CoA synthase and hydroxy methylglutaryl CoA reductase 1 were identified with more genes detected at 7 dpi than 3 dpi. A higher proportion of these genes were largely up-regulated, 34 out of 48 genes (Appendix A). Studies show that isoprenoids have various key roles in plant physiology including synthesis of key phytohormones (cytokinins, abscisic acid, gibberellins, and brassinosteroids) as well as plant defense compounds [46]. Taken together, we can hypothesize that *M. javanica* alters isoprenoid biosynthetic pathway to modulate hormone signaling which determines the outcome of plant–nematode interaction as reviewed by Gheysen and Mitchum [47]. Furthermore, up-regulation of genes related to secondary metabolism suggests mounting of a defense response as a result of damage caused by *M. javanica* feeding.

### 3.3. Nematode Responsive Phytohormones and Transcription Factors

Plant hormones are key players in mediating plant defense following nematode attack. Similarly, in this study *M. javanica* infestation influenced the expression of several genes associated with the synthesis and signaling of JA, SA, ET, auxin, abscisic acid (ABA), and brassinosteroids (BR) (Figure 4 and Figure 6).

The salicylic acid pathway is effective against biotrophic parasites while JA and ET pathway synergy is associated with enhanced resistance against necrotrophic microbes and herbivorous insects [48]. In this study, 9 out of 12 genes associated with SA biosynthesis were up-regulated (Figure 6A). Moreover, genes encoding for PR-1 (five genes) a robust marker of SA responsive genes expression were up-regulated at 3 and/or 7 dpi (Figure 5D and Appendix A), indicating activation of systemic induced resistance. Genes encoding for enzymes involved in the JA synthesis allene oxide synthase (*AOS* 2 genes), allene oxide cyclase (*AOC*, one gene), lipoxygenase (*LOX*, four genes), and 12-oxophytodienoate (*12-OPR*, three genes) were down-regulated at both time points (Figure 6B and Appendix A). Consequently, down-regulation of *LOX* and *12-OPR* enzymes might have played a role in initiating a susceptible interaction through interfering with JA-mediated defense pathway. The *LOX* pathway mediates resistance against phytopathogens including nematodes [49]. Plants incapable of producing JA or 12-oxo-phytodieonoic acid (*OPDA*) are more susceptible to phytonematodes [50]. Moreover, four protease inhibitors coding genes were down-regulated in this study at 3 and 7 dpi (Figure 5D and Appendix A). Jasmonic acid enhances expression of protease inhibitors which prevent proteolytic activity of the insect’s enzymes to debilitate their growth and reproduction. Nematodes essentially, depend on proteases to acquire nutrients as their source of food [47]. Hence, it is likely that JA plays a role in mediating defense against *M. javanica*. The antagonistic effect of SA on the JA pathway is often expressed at the gene transcription level, where SA targets the JA signaling. For instance, *WRKY50* plays a role in SA mediated suppression [48] was up-regulated in the current study (Figure 7C and Appendix A). Furthermore, TGA, sub-class of basic leucine zipper (bZIP) family of TFs are important molecular players in mediate JA suppression through SA. In this study, two genes coding for *TGA4* were up-regulated (Appendix A) further signifying suppression of JA pathway through induction of SA signaling pathway. Taken together, we can deduce that *M. javanica* adopts the strategy of targeting the SA pathway which concurrently suppresses JA-dependent defense responses resulting in potato susceptibility.

In addition to suppression of JA defense pathway, genes involved in ET synthesis, 1-aminocyclopropane-1-carboxylate synthase, and 1-aminocyclopropane-1-carboxylate were differentially regulated in this study with 4 and 5 genes down-regulated, respectively (Figure 7C and Appendix A). We also detected a gene encoding for ETHYLENE RESPONSE SENSOR 1 up-regulated (Figure 7C and Appendix A) which acts as negative regulators of ET responses suggesting ET suppression. Previous research demonstrated that ET inhibits RKN infestation, probably through a decrease in nematode attraction to the roots [47]. In the same way, rice resistant plants show more up-regulation of ET biosynthesis and response genes than susceptible plants infested with RKN [14,51].

Transcriptional reprogramming is a key hallmark of plant innate immunity. Members of several families of TFs (WRKY, bZIP, bHLH, NAC, MYB, and AP2/ERF) are well-known in regulating plant immune response [52]. In this study, we detected differential regulation of genes encoding for ERF, WRKY, MYB, bZIP, and DOF family of TFs among others (Figure 7A). Classification and identification of the differentially expressed TFs was attained from the Plant Transcription Factor Database (http://planttfdb.cbi.pku.edu.cn/ v.4.0) [53]. Most of the differentially expressed TFs in the current data set were down-regulated (298/532) after nematode infestation (Appendix A). The importance of TFs in mediating potato disease resistance to *Pectobacterium brasiliense 1692* and *Phytophthora infestans* infestation has been reported previously [17,18]. This is an indication that TFs are key regulators of potato immune response to various biotic stresses.

In this study, most of the genes (70.37%) encoding for AP2/ERF TF family were down-regulated at both 3 and/or 7 dpi. In addition, 15 genes were suppressed to a greater extent at 7 than 3 dpi (Figure 7B and Appendix A).

This could be ascribed to the secretion of nematode effectors and subsequent suppression of defense response associated with the activation of AP2/ERF TFs. The ERF subfamily of AP2/ERF TFs are important transcriptional regulators of genes responsive to biotic stress, particularly genes related to the JA and ET hormone signaling pathway [52]. Interestingly, we found three genes encoding for ERF6 up-regulated following *M. javanica* infestation (Figure 7B and Appendix A). This could imply that ERF6 TFs have a negative role in mediating potato susceptibility to *M. javanica*. Furthermore, seven genes coding for DREB were down-regulated in this study (Figure 7B and Appendix A).

It is generally accepted that pathogen-directed modulation of WRKY genes in plants is an important aspect that enhances success rates of pathogen infestation. Cyst nematode’s successful infestation process in *A. thaliana* roots was attributed to the nematode’s control over the expression of WRKY genes [54]. In agreement with this notion, we found 23 genes down-regulated WRKY-encoding genes, including *WRKY40, WRKY23, and WRKY29* both infection stages (Figure 7C and Appendix A). The MYB TFs have been characterized as an important regulator of both biotic and abiotic stresses [55]. Among the 34 down-regulated MYB TFs in this study, we found genes encoding for *MYB108* (3 genes), *MYB105,* and *MYB 14* at 3 and 7 dpi (Figure 7D and Appendix A).

We also identified genes encoding for DNA-binding with one finger (DOF) TFs (14 out 17 genes up-regulated (Figure 7E and Appendix A), transcriptional regulators acting on the formation and function of vascular tissues [56]. This further highlights the importance of vascular-related genes in establishment and formation of NFSs, therefore successful *M. javanica* colonization. Overall, the large number of differentially expressed TFs in this study reflects the complexity associated with plant defense regulation which orchestrates plant immunity leading to physiological alteration of the host to favor *M. javanica* infestation.

### 3.4. Cell Wall Organisation and Transport Processes Regulation by Meloidogyne javanica

This study showed that genes related to cell wall synthesis, degradation, modification, and cell wall proteins were differentially expressed following nematode attack (Figure 8A and Appendix A).

In addition, genes encoding for hydrolytic enzymes involved in pectin degradation such as polygalacturonase (PG), pectate lyases (PL), and pectin esterase (PE) were differentially expressed at 3 and 7 dpi by nematode infestation (Appendix A). Further, genes encoding for pectin methylesterase inhibitor (PMEI) were largely down-regulated (10 out of 12 genes) (Figure 8A and Appendix A). More PMEI genes were suppressed at 7 dpi compared to 3 dpi. Repression of PMEI by nematode attack shows that the activity of PME was activated leading to the breakdown of pectin bonds, increasing the vulnerability of the cell wall to microbial pectic enzymes, and other degrading enzymes, culminating to a susceptible response. We also detected xyloglucan endotransglucosylase/hydrolase and expansin encoding genes, cell wall modifying enzymes under differential regulation following nematode attack. Of these, 19 out of 21 genes encoding for xyloglucan endotransglucosylase/hydrolase were suppressed. Two genes (PGSC0003DMG400004670 and PGSC0003DMG400021877) were down-regulated at 3 dpi were up-regulated at 7dpi. Moreover, 4 genes encoding for expansin and expansin-like were precisely up-regulated at 7 dpi (Figure 8A and Appendix A). Xyloglucan endotransglucosylase/hydrolase and expansin enzymes have cell wall loosening properties required for plant cell wall expansion during plant development. Therefore, we can hypothesis that *M. Javanica* induces cell wall modifying enzymes at 7 dpi for expansion and enlargement of GCS a crucial event for nematode development. Further, we identified the alternation of genes related to beta-glucanase, 18 out of 29 genes down-regulated during disease progression (Figure 8A and Appendix A). Plant cell wall modification, the deposition of callose a 1,3-β glucan cell wall polymer, is associated with cell wall thickening at the site of pathogen attack acting as a physical barrier derailing further pathogen invasion [57]. Therefore, suppression of beta-glucanase genes is probable to interfere with plant cell wall reinforcement to ward off further nematode invasion, hence successful *M. javanica* colonization.

With the increased demand for nutrients in nematode feeding cells, nematodes deploy specialized membrane transporters to control the flow of nutrients in and out of the NFS [58]. Several genes encoding for peptide transporters (31 genes), major intrinsic proteins (22 genes), amino acid transporters (32 genes), metabolite transporters (27 genes), and sugar transporters (16 genes) were differentially expressed. Overall, we found that 55.29% of transporter encoding genes in the DEGs were up-regulated following nematode infestation (Figure 8B and Appendix A). The activation of genes encoding amino acid transporters (17 genes) and sugar transporters (7 genes) indicates regulation of amino acid and carbohydrate metabolism, respectively. Similarly, the induction of sugar transporters increases soluble sugar content in RKN infested tomato plants, which is crucial for nematode development [59]. Furthermore, multidrug transporter-encoding genes were differentially expressed in our samples which encompasses ATP- binding cassette (ABC, 42 genes) and multidrug and toxin extrusion proteins (MATE, 17 genes) (Figure 8B and Appendix A). These results may suggest that *M. javanica* recruits some of these transporters to flush out toxic secondary metabolites or to disperse nematode effectors produced during the infestation process.

### 3.5. Regulation of Proteolysis Processes by Meloidogyne javanica

Gene Ontology enrichment analyses showed that genes involved in primary metabolism were overrepresented among the down-regulated genes (Appendix A). Protein degradation was among the significantly altered processes involved in metabolism. In this study, genes coding for various classes of proteolytic enzymes were identified as differentially expressed include cysteine proteases, aspartate proteases, AAA-type, metacaspase, metalloprotease, subtilases, and serine proteases. The immense turnover of cellular proteins during nematode feeding can enhance a compatible nematode interaction due to regulation of plant immunity associated with proteolysis process.

Further, we found several genes involved in ubiquitin-proteasome system (UPS), the main protein degradation pathway in a cell. Various classes of E3 ubiquitin ligases enzymes that play a central role during protein ubiquitination process were differentially expressed. This includes SKP1-CUL1-F-box-protein (SCF) and the U-box domain RING finger protein-encoding genes which were co-regulated, 102 down- and 105 genes up-regulated (Appendix A). The fact that the UPS regulates degradation of many proteins in the cell affecting various cellular processes such as signal transduction, hormone signaling, and immune responses [60], makes it an attractive target for pathogen virulence factors including nematode effectors [61,62].

Collectively, this study uncovers the molecular networks regulated during compatible interaction between potato and *M. javanica*. Our results lay a foundation for functional studies in the future for genes highlighted to have a role in mediating susceptibility to *M. javanica* infestation to reveal their precise role in this interaction. In addition to providing further insights on plant–nematode interactions, further studies in the area development of target-specific control strategies against *Meloidogyne* species will originate.

## Figures and Tables

**Figure 1 microorganisms-08-01443-f001:**
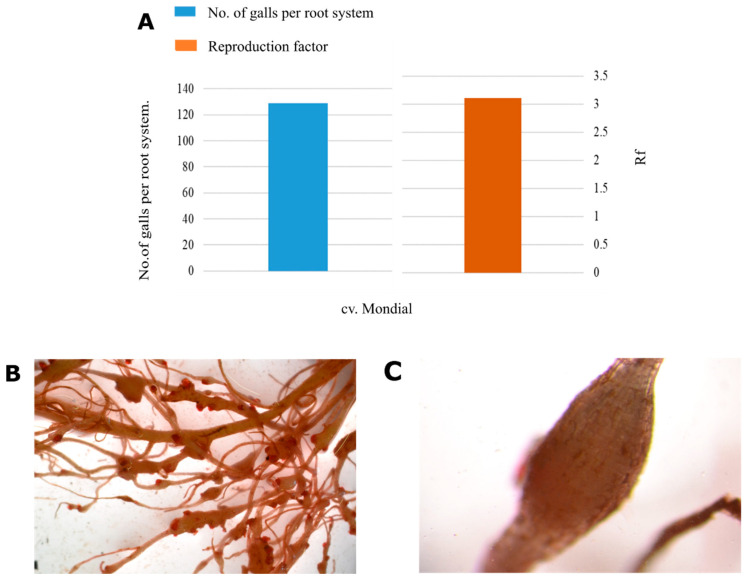
Responses of *Solanum tuberosum* cv. Mondial potato cultivar to *Meloidogyne javanica* infestation. (**A**) Reproductive factor and the number of galls induced by *Meloidogyne javanica* (**B**) and (**C**) displays nematode damage on potato roots, the egg masses stained pink and a mature gall, respectively.

**Figure 2 microorganisms-08-01443-f002:**
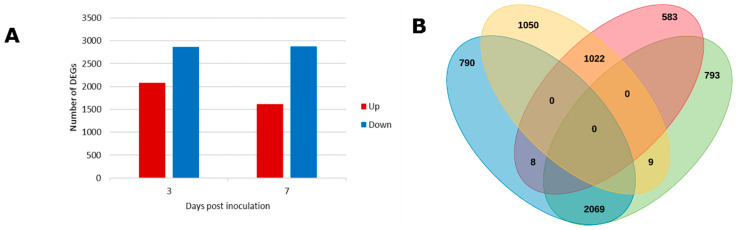
Schematic representation of differentially expressed genes in potato roots following *Meloidogyne javanica* infestation (**A**) Number of differentially expressed genes detected at 3 and 7 dpi compared to the mock-inoculated samples. ‘Down’ designates down-regulated genes. ‘Up’ designates up-regulated genes. (**B**) Venn diagram of the distribution of differentially expressed genes between 3 and 7 dpi. Yellow and blue ovals represent up-regulated and down-regulated differentially expressed genes at 3 dpi, respectively. Red and green ovals indicate the genes up-regulated and down-regulated at 7 dpi, respectively.

**Figure 3 microorganisms-08-01443-f003:**
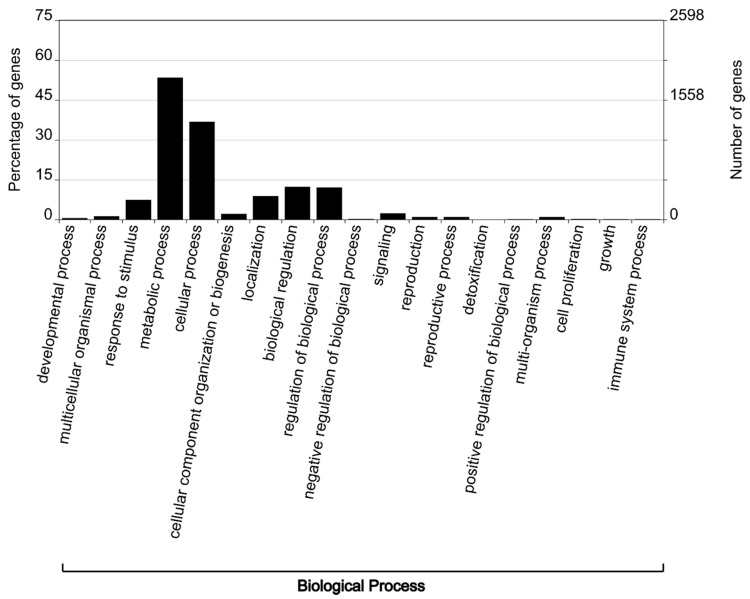
A representation of Gene Ontology analysis demonstrates the percentage of differentially expressed genes enriched within the biological process category.

**Figure 4 microorganisms-08-01443-f004:**
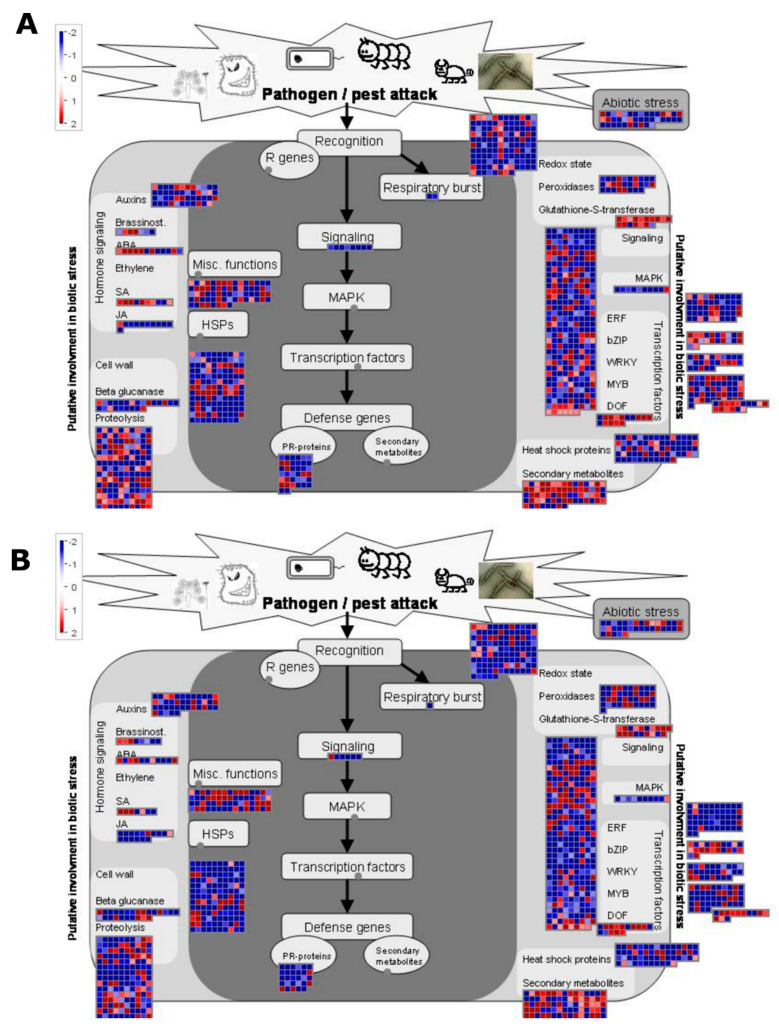
Transcriptional changes during responses of *Solanum tuberosum* cv. Mondial to *Meloidogyne javanica* infestation. ManMap visualization of genes related to biotic stress responses at 3 dpi (**A**) and 7 dpi (**B**). The modulation of 1471 and 1408 genes at 3 and 7 dpi, respectively, represents a set of genes important for early *Meloidogyne javanica* parasitism. The color-scale shown on the top left: red, up-regulated, and blue, down-regulated depicts the log_2_ fold-change of gene expression signal. The grey filled circle represents functional classifications to which no genes were mapped.

**Figure 5 microorganisms-08-01443-f005:**
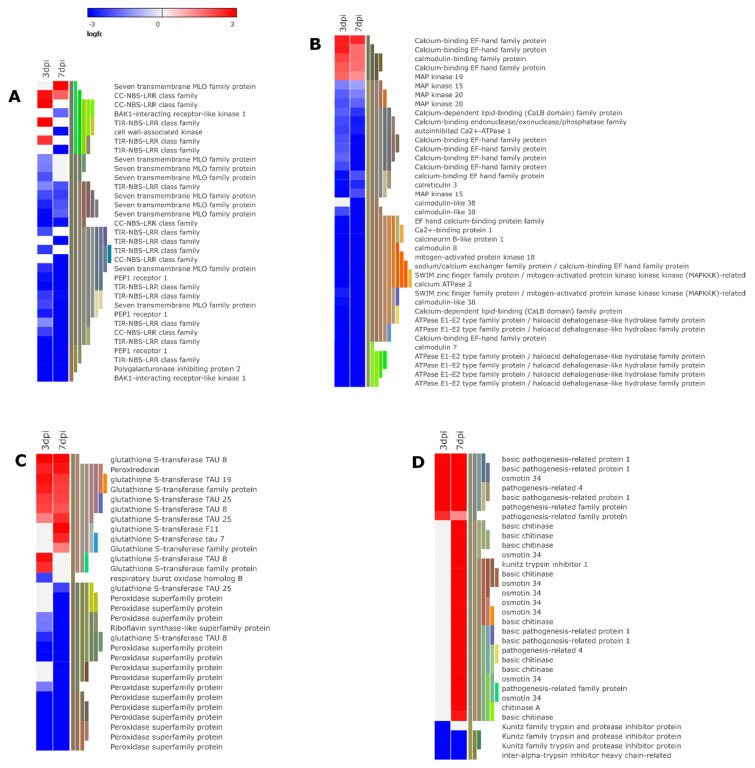
Heat map representation of selected differentially expressed genes associated with pathogen perception (**A**). Mitogen-activated protein kinase (MAPK) signaling pathways and calcium signaling pathways (**B**). Oxidative stress (**C**). Pathogenesis-related proteins and proteinase inhibitors (**D**). (The heat map illustrates a subset of genes from each group. Refer Appendix A for all differentially expressed genes in each group).

**Figure 6 microorganisms-08-01443-f006:**
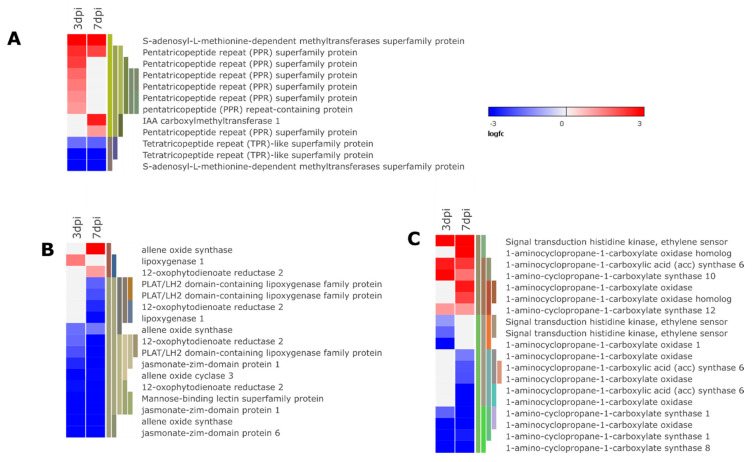
Heat map illustration of differentially expressed genes involved in hormone signal transduction. (**A**) Salicylic signaling pathway. (**B**) Jasmonic signaling pathway. (**C**) Ethylene signaling pathway. (The heat map illustrates a subset of genes from each group. Refer Appendix A for all differentially expressed genes in each group).

**Figure 7 microorganisms-08-01443-f007:**
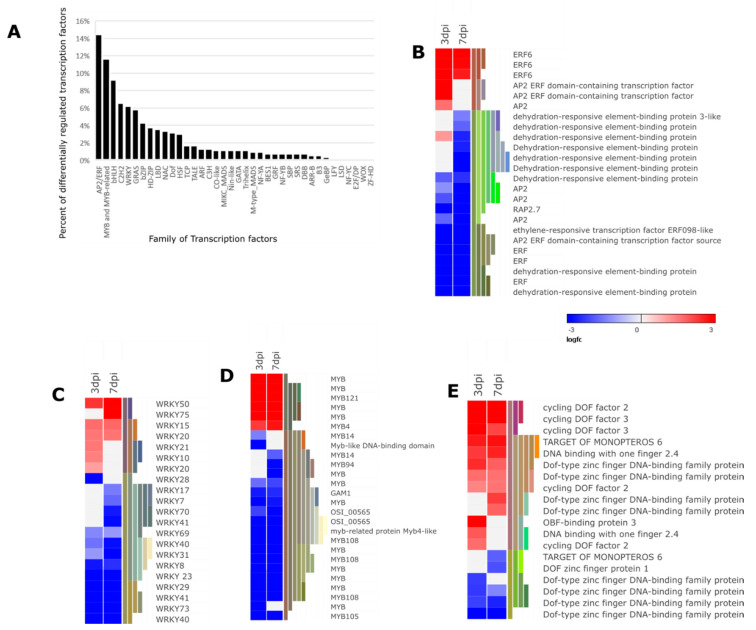
Heat map representation of differential regulation of transcription factors. (**A**) Represents various families of TFs under the regulation of *Meloidogyne javanica*. (**B**) AP2/ERF family. (**C**) WRKY family. (**D**) MYB family. (**E**) DOF family (The heat map represents a subset of the differentially expressed family of TFs. Refer to Appendix A for all transcription factors family displaying differential expression).

**Figure 8 microorganisms-08-01443-f008:**
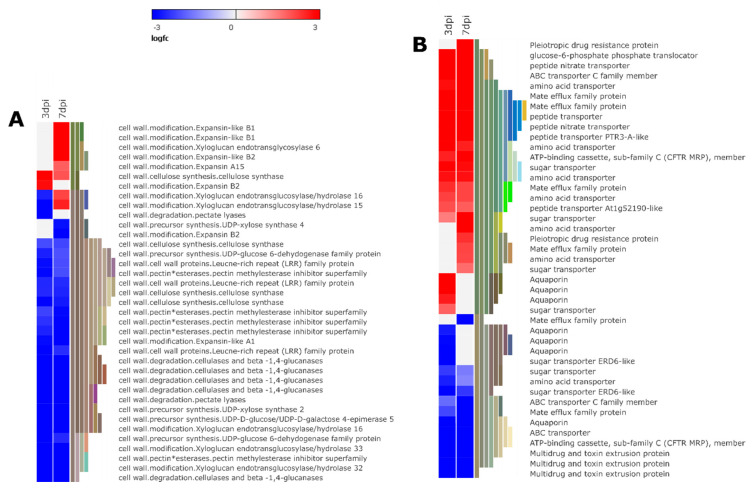
Heat map representation of gene expression patterns of genes associated with cell wall architecture (**A**) and transport activity (**B**). (The heat map illustrates a subset *of* genes from each group. Refer to Appendix A for all differentially expressed genes in each group).

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
