# Peer review of "Transcriptome Profiling of Potato (Solanum tuberosum L.) Responses to Root-Knot Nematode (Meloidogyne javanica) Infestation during A Compatible Interaction"

_microorganisms, 2020, doi:10.3390/microorganisms8091443_

Round 1

Reviewer 1 Report

This study investigated transcriptional responses of potato roots during a compatible interaction with root-knot nematode. The study contains new  elements and it can be considered publication after suitable revisions.

Suggestions:

L16: cv in full: first mention. cultivar (cv.)

L24: jasmonic acid

L39: root-knot nematodes (RKNs)

L79: S. tuberosum

L92: Sasser et al. - check throughout the text

L166: Fig 1. Solanum tuberosum cv.; Meloidogyne javanica

L189: Fig 2. Meloidogyne javanica

L212: Give in full GO and DEGs

L338: Give in full DEG, SA, JA, ET

L388: Fig 7A: X and y axis have no titles.

References:

Journal names are inconsistent. Check them through the whole section.

Check all latin names. They should be in italic. Some examples:

L540: Musa acuminata - in italic

L541: Meloidogyne incognita - in italic

L569: Solanum tuberosum - in italic

L577: Solanum sparsipilum - italic 

Author Response

16: cv in full: first mention. cultivar (cv.)

Thank you for noting this. We have corrected this by writing the cv in full after the first mention

L24: jasmonic acid

Thank you for noting this. We have addressed this by replacing capitalized J in Jasmonic with lowercase

L39: root-knot nematodes (RKNs)

Thank you for noting this. We have added the abbreviation for root-knot nematodes (RKNs)

L79: S. tuberosum

Thank you for noting this. We have addressed this (Line 96)

L92: Sasser et al. - check throughout the text

We have reviewed all the citation and references to ensure they are in line with journal specifications

L166: Fig 1. Solanum tuberosum cv.; Meloidogyne javanica

Thanks for noting this. We have addressed this by writing the scientific names in full (Lines 214-215)

L189: Fig 2. Meloidogyne javanica

Thanks for noting this. We have addressed this by writing the scientific names in full (Lines 255-260)

L212: Give in full GO and DEGs

Thanks for noting this. We have addressed this by replacing the abbreviated words with the full words (Lines 290-291)

L338: Give in full DEG, SA, JA, ET

Thanks for noting this. We have written in full the abbreviated words in the figure legend  (Lines 554-557)

L388: Fig 7A: X and y axis have no titles.

As suggested, we have included a new figure indicating both X and Y in figure 7A. (Line 639)

References:

Journal names are inconsistent. Check them through the whole section.

We agree with your assessment, we have counter checked all the names of the journals and addressed the inconsistency.

Check all Latin names. They should be in italic. Some examples:

L540: Musa acuminata - in italic

L541: Meloidogyne incognita - in italic

L569: Solanum tuberosum - in italic

L577: Solanum sparsipilum - italic 

As proposed, all the Latin names in the reference section have been italicized to ensure consistency in this reference section

Reviewer 2 Report

I found the article “Transcriptome profiling of potato (Solanum tuberosum L.) responses to root-knot nematode (Meloidogyne javanica) infection during a compatible interaction” to be an interesting approach to give in-depth information about the transcriptional responses of potato roots after M. javanica infestation. The authors have prepared a well-written manuscript, presenting sufficient data as supplementary material to further explain their results.

I have some minor points that need to be addressed:

I found the results/discussion part extremely long. The authors try to explain every single result and this complicates the reading of the manuscript a lot. They should consider reducing the size of this section

The authors in their text must be more precise, as they keep referring to RKN infestation, thus generalizing their results, and not to M. javanica. Even though, M. javanica is a root-knot nematode, such generalization of the results should be avoided. Their results do not allow them to do so.

The authors should consider replacing “infect” with “infest “, as well as “infection” with “infestation” throughout their text.

In M&M a new section entitled statistical analyses must be added. In the results, the authors present in several parts the type of analysis that they used (e.g. lines 180-182, 203-294), but all this must be together in a single section.

I found figure 4 to be very complexed and the legend is not providing sufficient information to explain it further to the reader.

Also, in all figure legends, the use of abbreviations without explaining their meaning should be avoided.

Line 13: place M. javanica out of the brackets  

Line 14 please replace "report" with "study"

Line 45. Please explain dpi as it is the first time mentioned in the text

Author Response

I found the results/discussion part extremely long. The authors try to explain every single result and this complicates the reading of the manuscript a lot. They should consider reducing the size of this section

We appreciate your insightful comments and suggestions, we have considered reducing the size of the results and discussion section and removed the following parts of the original draft (246-250, 274-277, 279-285, 309-394,403-409, 412-423, 431-435,447-482,500-502). We hope these edited sections provide more thorough and precise results and discussion.

The authors in their text must be more precise, as they keep referring to RKN infestation, thus generalizing their results, and not to  javanica. Even though M. javanicais a root-knot nematode, such generalization of the results should be avoided. Their results do not allow them to do so.

We agree with you. We have incorporated this suggestion throughout our results section by replacing RKN with Meloidogyne javanica making the study more specific. For instance, Line 15,16,20,30,31,201,223,267,387,416,419,544,547,550 and 591

The authors should consider replacing “infect” with “infest “, as well as “infection” with “infestation” throughout their text.

As suggested, we have considered replacing the word infect with infest and infection and infestation through the text. For instance, Line 4, 24, 28, 97, 105, 190, 224  231, 270, 409,419,550,606,614

In M&M a new section entitled statistical analyses must be added. In the results, the authors present in several parts the type of analysis that they used (e.g. lines 180-182, 203-294), but all this must be together in a single section.

As suggested, we have added a section on statistical analysis. (Materials and methods section- Subheading 2.7, Line 196-199)

I found figure 4 to be very complexed and the legend is not providing sufficient information to explain it further to the reader.

We agree with your assessment. We have rewritten the legend on figure 4 to give more precise information on what we attempted to convey (Line 304-309)

Also, in all figure legends, the use of abbreviations without explaining their meaning should be avoided.

Thank you for noting this. We  have corrected this by writing all the abbreviation in figure legends in full for instance, Lines 214-215,255-260, 290-291,554-557

Line 13: place  javanica out of the brackets

Thank you for noting this. We have placed the name Meloidogyne javanica out of the brackets

Line 14: please replace "report" with "study"

As suggested, we have replaced the word report with study

Line 45. Please explain dpi as it is the first time mentioned in the text

Thank you for noting this. However, we have already explained the meaning of dpi at first mentioning at line 17

Round 2

Reviewer 2 Report

The authors have addressed well all my comments. The current version of the manuscript is very improved. If possible, the authors must increase the resolution and the size of the graphs